# Does weight mediate the effect of smoking on coronary heart disease? Parametric mediational g-formula analysis

Yaser Mokhayeri[1], Maryam Nazemipour[2], Mohammad Ali Mansournia[3]*, Ashley I. Naimi[4], Jay S. Kaufman[5]

1 Cardiovascular Research Center, Shahid Rahimi Hospital, Lorestan University of Medical Sciences, Khorramabad, Iran, 2 Osteoporosis Research Center, Endocrinology and Metabolism Clinical Sciences Institute, Tehran University of Medical Sciences, Tehran, Iran, 3 Department of Epidemiology and Biostatistics, School of Public Health, Tehran University of Medical Sciences, Tehran, Iran, 4 Department of Epidemiology, Emory University, Atlanta, GA, United States of America, 5 Department of Epidemiology, Biostatistics and Occupational Health, Faculty of Medicine, McGill University, Montreal, Quebec, Canada

* mansournia_ma@yahoo.com

**Data Availability Statement:** The data underlying this article were provided by the National Heart,

## Abstract

### Background

In settings in which there are time-varying confounders affected by previous exposure and a time-varying mediator, natural direct and indirect effects cannot generally be estimated unbiasedly. In the present study, we estimate interventional direct effect and interventional indirect effect of cigarette smoking as a time-varying exposure on coronary heart disease while considering body weight as a time-varying mediator.

### Methods

To address this problem, the parametric mediational g-formula was proposed to estimate interventional direct effect and interventional indirect effect. We used data from the Multi-Ethnic Study of Atherosclerosis to estimate effect of cigarette smoking on coronary heart disease, considering body weight as time-varying mediator.

### Results

Over a 11-years period, smoking 20 cigarettes per day compared to no smoking directly (not through weight) increased risk of coronary heart disease by an absolute difference of 1.91% (95% CI: 0.49%, 4.14%), and indirectly decreased coronary heart disease risk by -0.02% (95% CI: -0.05%, 0.04%) via change in weight. The total effect was estimated as an absolute 1.89% increase (95% CI: 0.49%, 4.13%).

### Conclusion

The overall absolute impact of smoking to incident coronary heart disease is modest, and we did not discern any important contribution to this effect relayed through changes to

Lung, and Blood Institute (NHLBI) by permission Research Materials Distribution Agreement (RMDA) V02 1d20120806. Data will be shared with permission of NHLBI (https://biolincc.nhlbi.nih.gov/studies/mesa/). Please contact corresponding author or NHLBI for data access. Contact information: US Mail BioLINCC c/o Information Management Services, Inc. 3901 Calverton Blvd, Suite 200 Calverton MD 20705 Telephone 301-680-9770 - ask to speak with a BioLINCC representative Fax 301-680-8304 - please address fax communications to "BioLINCC" to ensure proper routing Email biolincc@imsweb.com.

**Funding:** The authors received no specific funding for this work.

**Competing interests:** The authors have declared that no competing interests exist.

bodyweight. In fact, changes in weight because of smoking have no meaningful mediating effect on CHD risk.

## Introduction

Coronary heart disease (CHD), also known as coronary artery disease and ischemic heart disease, is the leading cause of death in the world [1]. Although, mean percentage change in number of Years of Life Lost (YLL) from 2007 to 2017 due to CHD has increased by 17.3%, mean percentage change in age-standardized YLL rate has decreased by -9.8% in the same period [2].

Exposure to smoking between 1990 and 2015 declined worldwide by 25%; nevertheless, it still ranked among the leading five risk factors for attributable disability-adjusted life year (DALYs). In high-income countries such as the US, Canada, and the UK, smoking is considered the most important risk factor for attributable DALYs for both sexes [3].

LDL cholesterol, HDL cholesterol, triglyceride, body mass index, glucose, and blood pressure may mediate the relationship between smoking and CHD [4]. In smokers compared to nonsmokers, body mass indices are lower [5–7]. Chen et al. (2019), reported that smoking cessation could reduce 10-year risk of CHD, however, gaining weight following smoking cessation could conceal to a small extent the beneficial effect of the quitting [8]. Tamura et al. (2010) concluded that in spite of gaining weight after smoking cessation, the total risk of CHD was decreased in men [9]. Luo et al. (2013) according to their study on women, indicated that the relationship between smoking cessation and incidence of CHD could be weakened by gaining weight [10]. None of these studies appropriately adjusted for time-varying confounders using cohort data with repeated exposure and mediators [11–13]. G-methods (generalized methods) as causal methods could appropriately adjust for time-varying confounders affected by the prior exposure [14–18]. In frameworks with longitudinal data, intermediate confounder could not be adjusted using standard analytic approaches [19]. The intermediate confounder, which is likely not rare in mediation analysis, is both a mediator-outcome confounder and a variable in the causal direct path [19]. Moreover, to identify natural direct effect (NDE) and natural indirect effect (NIE) four assumptions are needed namely 1- no unmeasured exposure-outcome confounder 2- no- unmeasured mediator-outcome confounder 3- no unmeasured exposure-mediator confounder, and 4- no mediator-outcome confounder affected by the prior exposure [20]. In the presence of intermediate confounder, the fourth assumption would be violated, consequently natural direct and indirect effect are not identified. Addressing partially this problem, VanderWeele and Tchetgen Tchetgen using the mediational g-formula sought to find a solution and proposed the pure interventional direct effect (IDE) also known as a random interventional analogue of the natural direct effect and total interventional indirect effect (IIE) [21]. As VanderWeels and Tchetgen Tcheten's method was semiparametric, Lin et al developed a fully parametric mediational g-formula [22].

In the present study, we estimate IDE and IIE of cigarette smoking—as a time-varying exposure—on CHD while considering weight as both a time-varying mediator of past cigarette smoking, and a time-varying confounder of future cigarette smoking, using the parametric mediational g-formula to control for time-varying confounding. We estimate the direct (not through weight) and indirect (through weight) effects in data obtained from the Multi-Ethnic Study of Atherosclerosis (MESA).

## Methods

### Study population

We used the Multi-Ethnic Study of Atherosclerosis (MESA) data, a community-based cohort of adults, started in 2000 on 6814 men and women aged 45 to 84 years. Some additional details of the study have been presented elsewhere [23].

Five participants were excluded from our analysis because they already had prevalent CVD diagnoses at baseline. At the baseline, there were 6809 participants. It should be noted, since mGFORMULA SAS macro does not handle missing data, the final analysis limited to those participants without death or loss to follow up during the period of study (4433 participants). The participants were followed for more than eleven years and five visits. The first visit was scheduled for July 15, 2000 through July 14, 2002 (24 months); the second visit, scheduled for July 15, 2002 through January 14, 2004 (18 months); the third visit, scheduled for January 15, 2004 through July 14, 2005 (18 months); the fourth visit, scheduled for July 15, 2005 through July 14, 2007 (24 months); and the fifth visit, scheduled for April 1, 2010 through September 30, 2011 (18 months).

### Exposure, mediator, outcome, and confounders

Average number of cigarettes smoked per day—as the exposure—was self-reported. The data were obtained via questionnaire. Participants were asked if they smoked during last 30 days, those who replied affirmatively were asked for report the average number of cigarettes smoked per day. In this study, we aimed to estimate the causal effect of 'had everyone been a smoker/ smoked 20 cigarettes per day' vs. 'had everyone been a non-smoker/smoked 0 cigarettes per day', over the course of follow-up, on incident CHD or death from CHD. We defined weight in kilogram (*kg*) as the mediator. Weight was assessed using a standard weighing scale to the nearest 0.5 *kg*. CHD events as the outcome included myocardial infarction, resuscitated cardiac arrest, definite angina, probable angina (if followed by revascularization), and CHD death. Questionnaires, interviews, and medical records were used to obtain the date of CHD to the nearest day. Time-varying covariates measured at all visits including intentional physical activity, total cholesterol, hypertension, hypertension medication, and current aspirin use entered in the models as potential confounders. Additionally, we also adjusted for baseline age, sex, race/ethnicity (White, Asian, Hispanic, and African-American), alcohol consumption (drinks per week), and diet score [24], baseline smoking (never, former, and current smoker), annual family income, education level. A standard and validated questionnaire was used to obtain data on recreational physical activity. Valid test kits were used to obtain data on total cholesterol. An appropriately sized cuff was used to obtain resting blood pressure from the right arm after five minutes in the seated position. Three readings were taken; the average of the second and third readings were used as the blood pressure levels in the study.

### Statistical analysis

If we describe *mediation* as a two-stage process: 1- *M-stage* which denotes the process that cause the mediator and 2- *Y-stage*, a process that cause the outcome; hence, four potential outcomes would be obtained. Y0M0 is the outcome that would be observed if an individual were unexposed and had a mediator if she/he would have if unexposed (Y0M0 = Y0). Y1M1 is the outcome that would be observed if an individual were exposed and had a mediator if she/he would have if exposed (Y1M1 = Y1). Y1M0 is the outcome that would be observed if an individual were exposed but had a mediator if she/he would have if unexposed. Y0M1 is the outcome that would be observed if an individual were unexposed but had a mediator if she/he

would have if exposed. Using the parametric mediational g-formula, the *M-stage* is a random draw form the distribution of the mediator among those with exposure status $A = a$. Therefore, it would have fixed the mediator to the level that is randomly chosen from the distribution of the mediator among those with exposure status. Mediational g-formula is related to both Robins' regular g-formula [12] and Pearl's mediation formula [25]. In fact, in the absence of mediation it reduces to g-formula and in the absence of time-varying confounders reduces to mediation formula. In defining outcomes this way, we are able to quantify outcome risks under several different exposure and mediator scenarios such that IDE and IIE could be estimated. We estimated $(\bar{a}_1, \bar{a}_2)$ for each $(\bar{a}_1, \bar{a}_2) = (\bar{a}, \bar{a})$, $(\bar{a}, \bar{a}^*)$, $(\bar{a}^*, \bar{a})$, $(\bar{a}^*, \bar{a}^*)$. These measures indicate the simulated risk of CHD. This way, pure IDE, total IIE, and interventional analogue of total effect (ITE) can be defined as Eqs 1, 2, and 3, respectively.

$$\text{IDE} = [Q(\bar{a},\ \bar{a}^*) - Q(\bar{a}^*,\ \bar{a}^*)] = [P(Y_{1M_0}) - P(Y_{0M_0})] \tag{1}$$

$$\text{IIE} = [Q(\bar{a},\ \bar{a}) - Q(\bar{a},\ \bar{a}^*)] = [P(Y_{1M_1}) - P(Y_{1M_0})] \tag{2}$$

$$\text{ITE} = [Q(\bar{a},\ \bar{a}) - Q(\bar{a}^*,\ \bar{a}^*)] = [P(Y_{1M_1}) - P(Y_{0M_0})] \tag{3}$$

Fig 1 is a causal directed acyclic graph [26–32] depicting the relationship between our time-varying exposure (cigarette smoking), time-varying mediator (weight), time-varying confounders, and CHD. $V$ denotes baseline confounders (e.g. age); $A$ represents the time-varying exposure (cigarette smoking); $M$ denotes time-varying mediator (weight); $L$ represents the time-varying confounders (e.g. physical activity) which are affected by the prior exposures; and $Y$ corresponds to the binary outcome (CHD). Subscripts of 0 and 1 correspond to visits 0 and 1 of the study, respectively (to simplify the graph, just 2 visits were depicted).

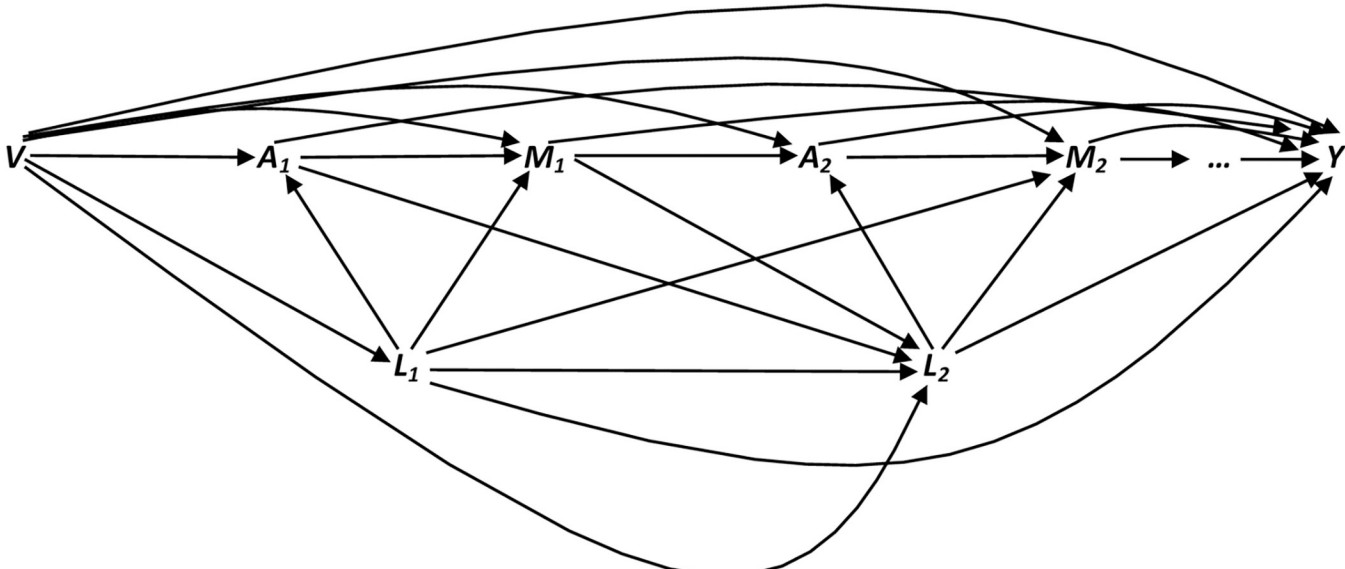

**Fig 1. Causal diagram depicting the effect of time-varying exposure (smoking) on the outcome (CHD) in the presence of time-varying mediator (weight).** V, A, M, L, and Y stand for baseline confounders, time-varying exposure, time-varying mediator, time-varying confounders, and the outcome, respectively.

We used Monte-Carlo estimation [33] to calculate the point estimates, and the nonparametric bootstrap with 500 resamples to obtain 95% confidence intervals. We used mGFORMULA SAS macro (S1 Appendix).

## Results

After 51,487 person-years of follow-up (median duration of follow-up 8.45 and IQR of 1.01 years) 388 new CHD cases occurred. Of 6809 eligible participants 401 (5.9%) died from causes other than CHD. Moreover, 1987 (29%) of participants were lost to follow-up during the study. Final sample included in the analysis is 4433 participants. Baseline characteristic of the 6,809 eligible participants are illustrated in Table 1. The smokers compared to quitters and nonsmokers were more likely to drink more alcohol, to have less intentional physical activity, to have less education levels, to have less family income, and to have less prevalence of hypertension and hypertension medication. The quitters compared to smokers and nonsmokers were likely to be men and to have more weight. Complete case analysis was performed, as the proportion of covariate missing data was low: For hypertension 2.2%, physical activity 1.5%, total cholesterol 3.1%, hypertension medication 3.2%, and current aspirin 1.6%.

The results of the standard parametric g-formula in Table 2 shows that the estimated 11-year risk of CHD in 'had everyone been a smoker/smoked 20 cigarettes per day' vs. 'had everyone been a non-smoker/smoked 0 cigarettes per day' were 6.92% and 5%, respectively. Therefore, risk ratio was estimated as 1.38 (95% CI: 1.04, 1.86). The estimated risk difference

**Table 1. Baseline characteristics of eligible participants in the Multi-Ethnic Study of Atherosclerosis, United States, 2000–2011.**

| Characteristics | Smoker (n = 890) | Quitter (n = 2493) | Nonsmoker (n = 3426) |
|---|---|---|---|
| **Age,** y, mean (SD) | 58.15 (9.15) | 63.48 (9.83) | 62.21 (10.52) |
| **Race/ethnicity,** n (%) | | | |
| Caucasian | 302 (33.93) | 1158 (46.45) | 1159 (33.83) |
| Asian | 45 (5.06) | 153 (6.14) | 606 (17.69) |
| African-American | 340 (38.20) | 697 (27.96) | 854 (24.93) |
| Hispanic | 203 (22.81) | 485 (19.45) | 807 (23.56) |
| **Male,** n (%) | 468 (52.58) | 1440 (57.76) | 1302 (38) |
| **Diet score,** mean (SD) | 2.50 (5.10) | 4.64 (4.77) | 4.61 (4.55) |
| **Alcohol consumption,** drinks/week, mean (SD) | 6.40 (11.32) | 5.81 (9.82) | 2.11 (5.81) |
| **Education,** n (%) | | | |
| ≤ high school | 350 (39.46) | 797 (32.09) | 1314 (38.48) |
| college–associate degree | 329 (37.09) | 761 (30.64) | 843 (24.69) |
| ≥ bachelor degree | 208 (23.45) | 926 (37.28) | 1258 (36.84) |
| **Annual family income,** n (%) | | | |
| < $25000 | 275 (32.51) | 666 (27.91) | 1115 (33.75) |
| $25,000-$50,000 | 277 (32.74) | 677 (28.37) | 938 (28.39) |
| > $50,000 | 294 (34.75) | 1043 (43.71) | 1251 (37.86) |
| **Hypertension,** n (%) | 333 (37.42) | 1178 (47.25) | 1545 (45.10) |
| **Hypertension medication,** n (%) | 265 (29.78) | 979 (39.27) | 1291 (37.68) |
| **Current aspirin use,** n (%) | 143 (16.07) | 601 (24.11) | 651 (19) |
| **Total cholesterol,** mmol/l, mean (SD) | 4.98 (1.01) | 4.98 (0.90) | 5.06 (0.91) |
| **Intentional physical activity,** min/week, mean (SD) | 351.93 (557.14) | 385.23 (547.45) | 354.64 (503.80) |
| **Weight,** kg, mean (SD) | 79.58 (16.92) | 81.81 (17.06) | 75.99 (16.93) |

SD: standard deviation.

**Table 2. Total effect of smoking 20 cigarettes per day compared to no smoking on CHD, using parametric g-formula in the Multi-Ethnic Study of Atherosclerosis, United States, 2000–2011.**

| Intervention | 11-year Risk, % | 95% CI | Risk Difference, % | 95% CI |
|---|---|---|---|---|
| No intervention | 5.94 | 3.74, 9.46 | 0.94 | 0.03, 2.01 |
| No smoking | 5.00 | 2.65, 8.60 | Ref. | |
| 20 cigarettes per day | 6.92 | 4.00, 11.14 | 1.92 | 0.19, 4.61 |

CHD: coronary heart disease, CI: confidence interval.

using standard parametric g-formula was 1.92%, which is close to the estimated risk difference for CHD—1.89%—using parametric mediational g-formula (Table 3).

In Table 3, using the estimates of the joint exposure and mediator interventions—simulated risks, we calculated interventional total effect, direct effect, and indirect effect (through weight) of smoking on CHD. Using parametric mediational g-formula we simulated CHD risks under no smoking with weight distributed as the weight under no smoking: *P(Y0M0)*; smoking 20 cigarettes per day with weight distributed as the weight under no smoking: *P(Y1M0)*; no smoking with weight distributed as the weight under smoking 20 cigarettes per day: *P(Y0M1)*; and smoking 20 cigarettes per day with weight distributed as the weight under smoking 20 cigarettes per day: *P(Y1M1)*. The results are presented on the risk difference scale. We estimated that smoking could directly (not through weight) increase CHD by 1.91% (95% CI: 0.49%, 4.14%), and decrease CHD as -0.02% (95% CI: -0.05%, 0.04%) via change in weight.

## Discussion

We found that smoking might have either a very small or no indirect effect on CHD through weight. Clair et al. (2013), testing the hypothesis that weight gain following smoking cessation does not attenuate the benefits of smoking cessation among adults with and without diabetes, indicated that weight gain after smoking cessation does not modify the association between smoking and risk of CVD events [34].

Feodoroff et al. investigating the dose-dependent effect of smoking on CHD in type 1 diabetes, indicated that smoking one pack per day (20 cigarettes) compared with never smokers adjusted for age, sex, BMI, hypertension, duration of diabetes, and $HbA_{1c}$ could increase the risk of incidence of CHD (HR = 1.45, 95% CI: 1.15, 1.84) [35]. Doyle et al. concluded that risk ratio for smoking 15–35 cigarettes per day compared to non-smokers in men is 1.6 [36]. Woodward et al. reported that men who smoke ≥20 cigarette per day compared to non-smokers for CHD have a hazard ratio 1.93 (95% CI: 1.15, 3.24). This value for women estimated as

**Table 3. Mediation analysis for the effect of smoking 20 cigarettes per day compared to no smoking on CHD, mediated by weight in the Multi-Ethnic Study of Atherosclerosis, United States, 2000–2011.**

| | Risk Difference, % | 95% CI |
|---|---|---|
| Interventional total effect | 1.89 | 0.49, 4.13 |
| Interventional direct effect | 1.91 | 0.49, 4.14 |
| Interventional indirect effect | -0.02 | -0.05, 0.04 |
| E (Y0M0) | 5.03 | 2.90, 7.95 |
| E (Y1M0) | 6.94 | 4.01, 10.3 |
| E (Y0M1) | 4.89 | 2.90, 7.91 |
| E (Y1M1) | 6.98 | 4.01, 10.05 |

CHD: coronary heart disease, CI: confidence interval.

3.81 (95% CI: 2.00, 7.27) [37]. Mucha et al. conducted a meta-analysis study and indicated that a low-level use of smoking (≤20 cigarettes per day) results in a risk ratio as 1.70 (95% CI: 1.52, 1.90) and a high-level use (>20 cigarettes per day) could cause a risk ratio of 2.09 (95% CI: 1.87, 2.34) [38]. Munafò et al. (2009), observed that at the baseline of a longitudinal study of men, never smokers and ex-smokers have a higher BMI (at average 1.6 kg/m2) compared to current smokers. The results were not changed after adjusting for age, socioeconomic position, alcohol, and calorie consumption. They also indicated an increase of 1.56 kg/m2 (95% CI: 1.29, 1.82) after smoking cessation adjusted for age, socioeconomic position, alcohol, and calorie consumption [5]. Sneve et al. using data of the Tromsø study resulted that current smokers in both genders compared to never- smokers have a lower BMI. They also reported that smoking cessation is related to an increase in weight [6].

The validity of the results depends on assumptions of positivity and consistency. The positivity assumption implies that the strata created by all confounder and treatment levels (exposed and unexposed) must be observed. However, the parametric g formula is less prone to bias induced by positivity violations. Consistency implies that the outcome for every exposed or unexposed individual (observed outcome) is equal with the outcome if they had received exposure or remained unexposed (counterfactual outcome), respectively. We note that similarity between observed and estimated risk is a necessary but insufficient condition for the assumption of no misspecification. In our study, the simulated 11-year risk of CHD under no intervention was 5.94% and the observed risk was 6.91%. As a strength of our analysis, the parametric mediational g-formula [22] is a valid statistical strategy to estimate the direct and indirect effects of time-varying exposure on incidence of CHD with time-varying mediator (weight). Using this statistical method, we could appropriately adjust for a variety of time-varying confounders, which are affected by prior exposures. As VanderWeele and Tchetgen Tchetgen's approach was semiparametric, we used a fully parametric mediational g-formula developed by Lin et al. (2017).

Our study is subject to some limitations. First, the validity of the estimated effects relies on the assumption of no unmeasured confounding for the effects of exposure on outcome, exposure on mediator and mediator on outcome, no measurement error, and correct specification of the parametric models used in the analysis. However, there was measurement error in self-reported cigarette smoking due to recall and underreporting biases cigarette smoking [39]. Physical activity was not assessed in visit 4, so we carried forward the values at visit 3. Moreover, we have not included occupational physical activity in the model. The presence of measurement error in confounders like physical activity and alcohol consumption will lead to residual confounding [40]. Second, the parametric mediational g-formula could not be used for multiple mediators. In fact, using this approach, we can just estimate one single direct/indirect effect. Third, the parametric-formula is also subject to the g-null paradox—it could lead to rejecting the causal null, even when it is true [41]. However, it could not be a concern in our setting as the effect of cigarette smoking and weight on CHD has been shown previously. Fourth, missing data, (censoring and competing risk) may result in selection bias but there is no function in the macro of mediational g-formula for handling missing data yet.

For future studies, Schomaker and Heumann (2018) regarding some estimators, which require bootstrapping to estimate confidence intervals, presented four methods to combine bootstrap estimation with multiple imputation. They indicated that the proportion of missingness and the number of multiple imputed data sets affect methods performance [42].

## Conclusion

The overall absolute impact of smoking to incident coronary heart disease is modest, and we did not discern any important contribution to this effect relayed through changes to

bodyweight. In fact, changes in weight because of smoking have no meaningful mediating effect on CHD risk.

## Supporting information

**S1 Appendix.**
(PDF)

## Acknowledgments

This manuscript was prepared using MESA Research Materials obtained from the National Heart, Lung, and Blood Institute (NHLBI); Research Materials Distribution Agreement (RMDA) V02 1d20120806. We would like to extend our thanks to the staff and coordinators of BioLINCC.

## Author Contributions

**Formal analysis:** Yaser Mokhayeri, Mohammad Ali Mansournia.

**Methodology:** Yaser Mokhayeri, Mohammad Ali Mansournia, Jay S. Kaufman.

**Software:** Maryam Nazemipour.

**Supervision:** Jay S. Kaufman.

**Validation:** Ashley I. Naimi.

**Writing – original draft:** Yaser Mokhayeri, Maryam Nazemipour, Mohammad Ali Mansournia.

**Writing – review & editing:** Mohammad Ali Mansournia, Ashley I. Naimi, Jay S. Kaufman.

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
