## [Decision Letter · Decision Letter 0]

7 Jul 2021

PONE-D-21-18594

Does Weight Mediate the Effect of Smoking on Coronary Heart Disease? Parametric Mediational G-Formula Analysis

PLOS ONE

Dear Dr. Mansournia,

Thank you for submitting your manuscript to PLOS ONE. After careful consideration, we feel that it has merit but does not fully meet PLOS ONE’s publication criteria as it currently stands. Therefore, we invite you to submit a revised version of the manuscript that addresses the points raised during the review process.

You are invited to further add the program codes in the supplementary files. Please also note that there might be attachments of the reviewers' comments.

We look forward to receiving your revised manuscript.

Kind regards,

Y Zhan

Academic Editor

PLOS ONE

Journal Requirements:

Reviewers' comments:

Reviewer's Responses to Questions

**Comments to the Author**

1. Is the manuscript technically sound, and do the data support the conclusions?

Reviewer #1: Partly

Reviewer #2: Partly

2. Has the statistical analysis been performed appropriately and rigorously? 

Reviewer #1: No

Reviewer #2: Yes

3. Have the authors made all data underlying the findings in their manuscript fully available?

Reviewer #1: No

Reviewer #2: Yes

4. Is the manuscript presented in an intelligible fashion and written in standard English?

Reviewer #1: Yes

Reviewer #2: Yes

5. Review Comments to the Author

Reviewer #1: PONE-D-21-18594: statistical review

SUMMARY. This paper describes a mediation analysis that evaluates whether weight mediates the effect of smoking on coronary heart disease. The core statistical analysis is based on a parametric mediational g-formula that has been recently proposed in the epidemiological literature. Results are in line with other studies that show factors that mediate smoking effects. The main problem with this paper is the presentation of the material, which is a bit too synthetic. Essentially, the authors simply state that they used a SAS macro and show the final results. It is therefore impossible to check whether the statistical methods are correct. Further, the results are not replicable. Technical soundness and replicability are two important requirements for publication in PLOSONE. I'd welcome a revision that adds details on the statistical part of the paper. See the list below for specific issues that should be addressed.

MAJOR ISSUES

1. Please clarify what parametric models have been fitted. Which covariates have been included? What was the structure of the random effects used to capture the longitudinal correlation of the data? Could the authors provide an evaluation of the goodness of fit of the model? Could the authors provide a traditional table with estimates and standard errors of the fitted parametric model?

2. Point estimates are computed by "non parametric bootstrap". Shouldn't parametric mediation analysis rely on simulation from the fitted parametric model? Why do we need nonparametric bootstrap here? A nonparametric approach to bootstrap is not obvious under a longitudinal setting becase we need to account for longitudinal correlation. This is why parametric approaches are usually preferred. Please clarify. Furthermore, Monte Carlo outputs are not replicable by definition. The authors should provide data and code (including the chosen random seed) to allow for replicability.

3. The authors say that "there is no function in the macro of mediational g-formula for handling missing data yet". Does it mean that only complete records have been used in the analysis? If it does, then this is a serious limitation that could invalidate the whole study. Please clarify.

Reviewer #2: Summary of the manuscript

Mokhayeri et al. aimed to explore what mediating effect changes in weight play in the effect of smoking on risk of coronary heart disease (CHD). As smoking status and weight are dynamic factors which can vary over time, they utilise the parametric mediational g-formula to unbiasedly account for exposure-caused mediator-outcome confounding in their analysis. Consistent with prior work they find smoking 20 cigarettes per day compared to none significantly increases the risk of CHD over time, though do not find strong evidence that changes in weight mediate these effects. Overall, this manuscript was a pleasure to read and reflects a well carried-out project with clear public health implications. It does however have a number of minor areas which could be improved prior to publication which would strengthen the manuscript. These mainly concern clarifying causal estimands of interest, methodological decisions and their rationale, providing a clearer explanation of how missing data was accounted for, and providing additional details in the results section.

Areas of improvement

Title and abstract:

The title and abstract are clear and accurately summarise the findings of the paper. As minor areas of improvement, however:

1. The final sentence of the abstract conclusion section may be better worded as 'changes in weight’ as a result of smoking do not appear to have an effect on CHD risk, instead of ‘losing weight’.

2. Likewise, while the summary conclusions are sound, it may be more appropriate to simply say changes in weight have no meaningful mediating effect on CHD risk instead of indicating a slight beneficial effect may be possible, given the 95% confidence interval could equally reflect a slight detrimental effect of weight on CHD risk (95% CI: -0.05%, +0.04%).

Introduction:

Overall, the introduction section is very well-done, providing a clear background and rationale to the project as well as justifying the use of the mediational g-formula in answering the specific research question of interest.

1. One recommended change however would be for authors to state which of the interventional direct and indirect effects estimated are pure and total (i.e. whether the additive interaction between exposure and mediator is incorporated into the ‘direct’ effect or the ‘indirect’ effect). While this can be worked out through the IDE and IIE formulas provided in pages 6 and 7, making each estimand clear could be helpful for readers given the slight difference in interpretation. For example, if the pure IIE was estimated (YA0M1 - YA0M0), this would reflect the effect of having weight similar to smokers (vs. non-smokers) on CHD risk in a population where everyone was a non-smoker. The total IIE on the other hand would reflect the effect of having weight similar to non-smokers (vs. smokers) in a population where everyone was a smoker, so they get towards different questions.

Methods:

While the methods section is well-done overall and provides ample information on the study design and setting including the time-periods of each visit, there are several areas where improvements could be made that in my view would strengthen the resulting paper.

1. Authors need to provide a description of how missing data is addressed, given its particular importance to the mGFORMULA SAS macro. If understood correctly, a complete case analysis was performed using only those who did not experience a competing risk event (5.9% of the 6,809 eligible participants) or right-censoring due to loss to follow-up (29% of participants). Given the mGFORMULA SAS macro cannot accommodate competing risks or censoring and requires that all participants included have an equal number of follow-up visits/time-points, this should be stated explicitly as well as the size of the final sample actually included in the analysis.

2. In the ‘Exposure, mediator, outcome, and confounders’ section of the methods, authors describe their comparison as a hypothetical intervention forcing all individuals to smoke 20 cigarettes per day compared to none and interpret it as such. As is discussed elsewhere in the literature though, such as by Schwartz, Gatto, and Campbell in the Annals of Epidemiology in 2016, this may not be the most appropriate interpretation of their comparison given the number of different ways this intervention could be enforced, each having a different relationship with future health and possibly violating Rubin’s Stable Unit Treatment Value assumption. Instead authors may wish to describe their comparison in terms of identifying the realised causal effect of smoking 20 cigarettes per day compared to none among participants in the sample who were exposed.

3. In the ‘Exposure, mediator, outcome, and confounders’ section of the methods, authors indicate that alcohol consumption, diet, and family income are included as time-fixed covariates based on baseline values. Given each of these factors are likely dynamic in nature and treating them as fixed could result in misclassification bias or residual confounding, authors should justify treating these factors as time-fixed, such as due to only having information available at baseline. If authors have longitudinal data on these factors, they should explore treating them as time-varying in their analysis or, if not, consider reporting the extent to which these factors vary within individuals in their sample throughout follow-up.

4. In the same section as above, authors indicate baseline smoking status was included as a time-fixed covariate. It would be helpful if it was made more explicit how this variable was accounted for. For example, if baseline smoking status was included as the average number of cigarettes smoked, it may not account for individuals being former smokers and the continuing CHD risk this would present. Likewise if it does not account for smoking history as of baseline, this would also fail to capture individuals who were former smokers still having a possibly greater risk of CHD than never-smokers. If possible, I would recommend authors include both current smoking status at baseline (as an average number of cigarettes smoked per day) as well as smoking history at baseline (as a categorical for current smoker, former smoker, never smoker) as time-fixed covariates.

5. In the ‘Statistical analysis’ section of the methods, I would recommend a different wording when discussing potential outcomes and exposure and mediator status’, making it explicit that each potential outcome refers to exposure and mediator histories over the course of follow-up instead of at a single time-point, with these histories being some exposure-mediator combination of {0,0,0,0,0} and/or {1,1,1,1,1}.

6. For the line ‘Mediational g-formula is both Robins’ regular g-formula and Pearl’s mediation formula’, I think it would be more appropriate to say the mediational g-formula is related to both Robin’s regular g-formula and Pearl’s mediation formula, only being equal under the conditions authors subsequently state.

7. Possibly in the ‘Statistical analysis’ section, authors should provide additional detail on whether any interaction terms are included between covariates such as between the exposure and mediator, and whether any non-linear terms are considered in the model specification for continuous factors, such as quadratic, cubic, or spline terms. This could be helpful for readers, in that these decisions could influence how accurately the parametric model captures the true causal model.

8. Authors should report any efforts made to address or explore possible biases. For example, authors could assess the validity of their parametric model specification through comparing the CHD risk simulated under the Natural Course (i.e. without intervention) to that actually observed in the sample. Likewise, authors could perform sensitivity analyses (though not required) excluding those who were former smokers so as to limit the likelihood of lagged effects of smoking before the intervention period. As a final suggestion, authors may wish to explore how their findings differ if analysed over a 9- or 13-year period to determine if their effect estimate is stable or affected by chance variation in year-to-year sample CHD incidence.

9. Though it could be a local issue, the Figure 1 causal DAG appears to have a low-resolution, making it difficult to read. If this is the case, authors may wish to provide a higher quality image for better legibility.

Results:

Authors present the findings of their analysis well, though there are minor areas where improvements could be made.

1. Related to point 1 in the ‘Methods’ improvements section, authors make clear that some individuals die from causes other than CHD or are lost to follow-up. Where possible, in the results section authors should report the final sample size included in the mediational g-formula analyses, as well as provide a brief summary of the characteristics of those not included in analyses to assess the likelihood of differential loss to follow-up by factors related to the outcome of interest.

2. Related to point 1 in the ‘Methods’ improvement section as well, authors do not make clear how missing data is accounted for. This is needed, given Table 1 race, education, and annual family income strata sizes do not sum to each smoking status group sample size (e.g. annual family income among smokers sum to 846, but there are 890 smokers at baseline), suggesting there is some missing data. Where possible, information on missing data should be provided for each variable separated by smoking status, as well as a complete case sample size carried forward for use with the mediational g-formula. This is relevant to the positivity assumption, where a smaller sample size is more likely to result in random non-positivity.

3. In summarising results of the standard parametric g-formula, authors describe the estimated 11-year risk of CHD in smokers and non-smokers. To reflect that the comparison is counterfactual, comparing outcomes had everyone been smokers or non-smokers, I would recommend they word this sentence differently as the estimated risk ‘had everyone been a smoker/smoked 20 cigarettes per day’ vs. ‘had everyone been a non-smoker/smoked 0 cigarettes per day’ for the 11-year follow-up.

4. In Tables 1 or 2, it would be helpful if authors presented the observed 11-year risk of CHD (ideally overall and by smoking status), as well as that simulated in the standard or mediational g-formula under no intervention. This would be a helpful assumption check, given the 'No intervention' output presented by the g-formula represents the simulated CHD risk under the Natural course (i.e. without intervention). If the parametric modelling process accounts for the causal model well, this 'no intervention' risk should be similar to that actually observed in the sample. If it isn't, it may suggest more work is needed in refining the model specification, such as considering different covariates, interaction terms, or non-linear terms for continuous covariates.

5. Either in Table 3 or separately, I would recommend authors present the simulated risk under each counterfactual exposure-mediator intervention. This would be helpful for readers, as it would allow multiplicative effects to be calculated such as E[YA1M1] / E[YA1M0] for the total IIE, as well as the interventional total direct effect and interventional pure indirect effect.

Discussion:

Overall, authors do an excellent job of summarising related research as well as noting strengths and limitations of their study. Some brief areas of possible improvements are listed below:

1. While authors summarise findings of a number of related articles, I feel the findings of these studies could be more explicitly compared against their own findings, such as through simply restating the summary total, direct, and indirect effects of their own study before discussing total, direct, and weight-mediated effect estimates found in other studies.

2. The discussion section could benefit from authors comparing effects of their analysis using the mediational g-formula to similar studies which did not utilise the mediational g-formula. For example, the 2013 JAMA study by Carole Clair et al. considers the mediating effect of weight in the relationship between smoking cessation and CVD among people with and without diabetes. This may be a more relevant comparison than the study by Nianogo and Arah where the outcome was type 2 diabetes mellitus and not directly related to CHD.

3. On page 11, authors suggest the parametric mediational g-formula is the only known method that is valid for investigating causal effects in the presence of time-varying mediation. This may not necessarily be correct, and could be adapted. As they note, VanderWeele and Tchetgen Tchetgen’s semiparametric approach is also valid. Further, the authors may find the 2017 paper by Zheng and van der Laan in the Journal of Causal Inference relevant, where they discuss the conditional mediation formula and multiple estimators for mediation effects in the presence of time-varying mediation through applying TMLE, IPW, or a non-targeted substitution estimator.

4. Authors currently split their discussion of positivity, consistency, and exchangeability/unmeasured confounding into two sections, separated by study strengths. The discussion may be better laid-out if kept together, relating each assumption to the evidence supporting it and detracting from it one at a time.

5. While authors appropriately note the g-null paradox as a limitation of the g-formula, if there is space in the manuscript it may benefit from a slightly longer discussion, especially given the authors note it is unlikely to be a concern in their study due to prior work suggesting the sharp null doesn’t hold in this context.

6. While authors do an excellent job reviewing study strengths and limitations, they may wish to review work published in Statistics in Medicine in 2018 by Schomaker and Heumann, where they discuss combining bootstrap inference with multiple imputation where the g-formula is used. While this would not account for competing risks and censoring, and is not being recommended as necessary in this review, missing data such as that mentioned in point 2 of the ‘results’ improvement section can be accounted for as a step prior to or alongside the mediational g-formula, either through storing each bootstrap effect estimate across multiply imputed datasets and using percentile-based confidence intervals, or through applying the mediational g-formula to multiply imputed datasets and pooling summary effect estimates through Rubin's rules. Each approach discussed is relatively straight-forward to implement in practice.

6. PLOS authors have the option to publish the peer review history of their article (what does this mean?). If published, this will include your full peer review and any attached files.

Reviewer #1: No

Reviewer #2: **Yes: **Kieran Blaikie

---

## [Author Response · Author response to Decision Letter 0]

11 Nov 2021

Reviewer 1

SUMMARY. This paper describes a mediation analysis that evaluates whether weight mediates the effect of smoking on coronary heart disease. The core statistical analysis is based on a parametric mediational g-formula that has been recently proposed in the epidemiological literature. Results are in line with other studies that show factors that mediate smoking effects. The main problem with this paper is the presentation of the material, which is a bit too synthetic. Essentially, the authors simply state that they used a SAS macro and show the final results. It is therefore impossible to check whether the statistical methods are correct. Further, the results are not replicable. Technical soundness and replicability are two important requirements for publication in PLOSONE. I'd welcome a revision that adds details on the statistical part of the paper. See the list below for specific issues that should be addressed.

MAJOR ISSUES

1. Please clarify what parametric models have been fitted. Which covariates have been included? What was the structure of the random effects used to capture the longitudinal correlation of the data? Could the authors provide an evaluation of the goodness of fit of the model? Could the authors provide a traditional table with estimates and standard errors of the fitted parametric model?

Our answer: Thank you. Considering our dichotomous outcome, the parametric model is based on a logistic regression and tries to estimate the standardized risk using the probability of confounders (both time-fixed and time-varying) and the conditional distribution of outcome (CHD) given the exposure (smoking) and set of confounders. Time-varying covariates measured at all visits including intentional physical activity, total cholesterol, hypertension, hypertension medication, and current aspirin use entered in the models as potential confounders. Additionally, we also adjusted for baseline age, sex, race/ethnicity (White, Asian, Hispanic, and African-American), alcohol consumption (drinks per week), and diet score. To assess the validity of the parametric model specification, we compared the CHD risk simulated under the Natural Course (without intervention) with the actually observed risk (added to the discussion). We reported CI using bootstrapping with 500 iterations. Regarding the structure of the random effects, there is no effect term in our model. The within subject-correlation has been taken into account using bootstrapping; and regarding the traditional table, in fact, the beta estimates of the outcome and covariate models and their standard errors are not our interest and so the software does not provide them.

2. Point estimates are computed by "non parametric bootstrap". Shouldn't parametric mediation analysis rely on simulation from the fitted parametric model? Why do we need nonparametric bootstrap here? A nonparametric approach to bootstrap is not obvious under a longitudinal setting because we need to account for longitudinal correlation. This is why parametric approaches are usually preferred. Please clarify. Furthermore, Monte Carlo outputs are not replicable by definition. The authors should provide data and code (including the chosen random seed) to allow for replicability.

Our answer: Indeed, we used the nonparametric bootstrap to obtain point estimates and confidence intervals for the effects of interest. Use of the nonparametric bootstrap is standard with the parametric g formula. In principle, the parametric bootstrap can also be used, but there is no requirement (i.e., one need not use parametric bootstrap with parametric g formula). The reason researchers employ the nonparametric bootstrap is that it is much simpler to implement, particularly in a setting of longitudinal data such as ours. Specifically, with a simple clustered resample, one can address the longitudinal correlations with the nonparametric bootstrap. Were we to rely on the parametric bootstrap, we would have to employ much more complex modeling strategies, which are not immediately compatible with the g formula.

The code—SAS macro—was added as a supplementary file. 

3. The authors say that "there is no function in the macro of mediational g-formula for handling missing data yet". Does it mean that only complete records have been used in the analysis? If it does, then this is a serious limitation that could invalidate the whole study. Please clarify.

Our answer: While parametric g-formula macro do consider censoring in the analysis, mgformula macro for parametric g-formula cannot handle censoring. Using mGORMULA we did a complete case analysis. We indicated this in the discussion. Final sample included in the analysis is 4433 (Censoring proportion: 34.9%).

Lin et al. (Epidemiology, 2016; and Statistics in Medicine, 2017) limited their analyses on participants without death or loss to follow up during the period of study. 

Reviewer 2

Mokhayeri et al. aimed to explore what mediating effect changes in weight play in the effect of smoking on risk of coronary heart disease (CHD). As smoking status and weight are dynamic factors which can vary over time, they utilise the parametric mediational g-formula to unbiasedly account for exposure-caused mediator-outcome confounding in their analysis. Consistent with prior work they find smoking 20 cigarettes per day compared to none significantly increases the risk of CHD over time, though do not find strong evidence that changes in weight mediate these effects. Overall, this manuscript was a pleasure to read and reflects a well carried-out project with clear public health implications. It does however have a number of minor areas which could be improved prior to publication which would strengthen the manuscript. These mainly concern clarifying causal estimands of interest, methodological decisions and their rationale, providing a clearer explanation of how missing data was accounted for, and providing additional details in the results section.

Areas of improvement

Title and abstract:

The title and abstract are clear and accurately summarise the findings of the paper. As minor areas of improvement, however:

1. The final sentence of the abstract conclusion section may be better worded as 'changes in weight’ as a result of smoking do not appear to have an effect on CHD risk, instead of ‘losing weight’.

Our answer: Thank you. It was revised. We made the following change: In fact, changes in weight because of smoking have no meaningful mediating effect on CHD risk.

2. Likewise, while the summary conclusions are sound, it may be more appropriate to simply say changes in weight have no meaningful mediating effect on CHD risk instead of indicating a slight beneficial effect may be possible, given the 95% confidence interval could equally reflect a slight detrimental effect of weight on CHD risk (95% CI: -0.05%, +0.04%).

Our answer: Thank you. It was revised. We made the following change: In fact, changes in weight because of smoking have no meaningful mediating effect on CHD risk

Introduction:

Overall, the introduction section is very well-done, providing a clear background and rationale to the project as well as justifying the use of the mediational g-formula in answering the specific research question of interest.

1. One recommended change however would be for authors to state which of the interventional direct and indirect effects estimated are pure and total (i.e. whether the additive interaction between exposure and mediator is incorporated into the ‘direct’ effect or the ‘indirect’ effect). While this can be worked out through the IDE and IIE formulas provided in pages 6 and 7, making each estimand clear could be helpful for readers given the slight difference in interpretation. For example, if the pure IIE was estimated (YA0M1 - YA0M0), this would reflect the effect of having weight similar to smokers (vs. non-smokers) on CHD risk in a population where everyone was a non-smoker. The total IIE on the other hand would reflect the effect of having weight similar to non-smokers (vs. smokers) in a population where everyone was a smoker, so they get towards different questions.

Our answer: Thank you. Indeed, it is a valuable recommendation. It was revised in both introduction and method sections. Interventional direct effect (IDE) is pure effect, and interventional indirect effect (IIE) is total effect.

Methods:

While the methods section is well-done overall and provides ample information on the study design and setting including the time-periods of each visit, there are several areas where improvements could be made that in my view would strengthen the resulting paper.

1. Authors need to provide a description of how missing data is addressed, given its particular importance to the mGFORMULA SAS macro. If understood correctly, a complete case analysis was performed using only those who did not experience a competing risk event (5.9% of the 6,809 eligible participants) or right-censoring due to loss to follow-up (29% of participants). Given the mGFORMULA SAS macro cannot accommodate competing risks or censoring and requires that all participants included have an equal number of follow-up visits/time-points, this should be stated explicitly as well as the size of the final sample actually included in the analysis.

Our answer: Thank you. Yes, mGFORMULA cannot handle competing risk and censoring, and using mGFORMULA we did a complete case analysis. We indicated this in the discussion. Final sample included in the analysis is 4433 (Censoring proportion: 34.9%) and this was added to the results section. 

Lin et al. (Epidemiology, 2016; and Statistics in Medicine, 2017) limited their analyses on participants without death or loss to follow up during the period of study. 

2. In the ‘Exposure, mediator, outcome, and confounders’ section of the methods, authors describe their comparison as a hypothetical intervention forcing all individuals to smoke 20 cigarettes per day compared to none and interpret it as such. As is discussed elsewhere in the literature though, such as by Schwartz, Gatto, and Campbell in the Annals of Epidemiology in 2016, this may not be the most appropriate interpretation of their comparison given the number of different ways this intervention could be enforced, each having a different relationship with future health and possibly violating Rubin’s Stable Unit Treatment Value assumption. Instead, authors may wish to describe their comparison in terms of identifying the realised causal effect of smoking 20 cigarettes per day compared to none among participants in the sample who were exposed.

Our answer: Thank you. It was revised and your suggestion was replaced in this section.

3. In the ‘Exposure, mediator, outcome, and confounders’ section of the methods, authors indicate that alcohol consumption, diet, and family income are included as time-fixed covariates based on baseline values. Given each of these factors are likely dynamic in nature and treating them as fixed could result in misclassification bias or residual confounding, authors should justify treating these factors as time-fixed, such as due to only having information available at baseline. If authors have longitudinal data on these factors, they should explore treating them as time-varying in their analysis or, if not, consider reporting the extent to which these factors vary within individuals in their sample throughout follow-up.

Our answer: Thank you. Information for diet and alcohol consumption (drinks per week) is available just at baseline. Regarding income, it was roughly stable over time so we include it as a time-fixed variable.

4. In the same section as above, authors indicate baseline smoking status was included as a time-fixed covariate. It would be helpful if it was made more explicit how this variable was accounted for. For example, if baseline smoking status was included as the average number of cigarettes smoked, it may not account for individuals being former smokers and the continuing CHD risk this would present. Likewise if it does not account for smoking history as of baseline, this would also fail to capture individuals who were former smokers still having a possibly greater risk of CHD than never-smokers. If possible, I would recommend authors include both current smoking status at baseline (as an average number of cigarettes smoked per day) as well as smoking history at baseline (as a categorical for current smoker, former smoker, never smoker) as time-fixed covariates.

Our answer: Thank you. We adjusted for smoking at baseline as a categorical variable (never, former, and current smoker). We made the following change: baseline smoking (never, former, and current smoker).

5. In the ‘Statistical analysis’ section of the methods, I would recommend a different wording when discussing potential outcomes and exposure and mediator status’, making it explicit that each potential outcome refers to exposure and mediator histories over the course of follow-up instead of at a single time-point, with these histories being some exposure-mediator combination of {0,0,0,0,0} and/or {1,1,1,1,1}.

Our answer: With all due respect, to the reviewer, we believe this definition is already simple and easy to understand, and we have made no further changes

6. For the line ‘Mediational g-formula is both Robins’ regular g-formula and Pearl’s mediation formula’, I think it would be more appropriate to say the mediational g-formula is related to both Robin’s regular g-formula and Pearl’s mediation formula, only being equal under the conditions authors subsequently state.

Our answer: Thank you. It was revised as suggested by the reviewer.

7. Possibly in the ‘Statistical analysis’ section, authors should provide additional detail on whether any interaction terms are included between covariates such as between the exposure and mediator, and whether any non-linear terms are considered in the model specification for continuous factors, such as quadratic, cubic, or spline terms. This could be helpful for readers, in that these decisions could influence how accurately the parametric model captures the true causal model.

Our answer: No interaction terms are included; however. Some non-linear terms are included. For cholesterol and weight, linear term was considered when used as dependent variables, and quadratic linear term when used as independent variables. For cigarette smoking and intentional physical activity logistic, then log-linear when used as dependent variable and quadratic linear term when used as independent variable. The code—SAS macro—was added as a supplementary file.

8. Authors should report any efforts made to address or explore possible biases. For example, authors could assess the validity of their parametric model specification through comparing the CHD risk simulated under the Natural Course (i.e. without intervention) to that actually observed in the sample. Likewise, authors could perform sensitivity analyses (though not required) excluding those who were former smokers so as to limit the likelihood of lagged effects of smoking before the intervention period. As a final suggestion, authors may wish to explore how their findings differ if analysed over a 9- or 13-year period to determine if their effect estimate is stable or affected by chance variation in year-to-year sample CHD incidence.

Our answer: Thank you for your suggestion. The natural course compared to the observed risk was added to the discussion. In our study, the simulated 11-year risk of CHD under no intervention was 5.94% and the observed risk was 6.91%. Regarding the analysis over 9- or 13-year period, it should be noted that the authors have no access to data of 13-year period. Moreover, in MESA, there is a gap of three years between visits of four and five. 

9. Though it could be a local issue, the Figure 1 causal DAG appears to have a low-resolution, making it difficult to read. If this is the case, authors may wish to provide a higher quality image for better legibility.

Our answer. The figure was replaced with a high-resolution one. 

Results:

Authors present the findings of their analysis well, though there are minor areas where improvements could be made.

1. Related to point 1 in the ‘Methods’ improvements section, authors make clear that some individuals die from causes other than CHD or are lost to follow-up. Where possible, in the results section authors should report the final sample size included in the mediational g-formula analyses, as well as provide a brief summary of the characteristics of those not included in analyses to assess the likelihood of differential loss to follow-up by factors related to the outcome of interest.

Our answer: final sample size was added. The authors assumed the missingness is completely at random (MCAR). 

2. Related to point 1 in the ‘Methods’ improvement section as well, authors do not make clear how missing data is accounted for. This is needed, given Table 1 race, education, and annual family income strata sizes do not sum to each smoking status group sample size (e.g. annual family income among smokers sum to 846, but there are 890 smokers at baseline), suggesting there is some missing data. Where possible, information on missing data should be provided for each variable separated by smoking status, as well as a complete case sample size carried forward for use with the mediational g-formula. This is relevant to the positivity assumption, where a smaller sample size is more likely to result in random non-positivity.

Our answer: Complete case analysis was performed, as the proportion of covariate missing data was low. For hypertension 2.2%, physical activity 1.5%, total cholesterol 3.1%, hypertension medication 3.2%, and current aspirin 1.6%.This was added to the result section. 

3. In summarizing results of the standard parametric g-formula, authors describe the estimated 11-year risk of CHD in smokers and non-smokers. To reflect that the comparison is counterfactual, comparing outcomes had everyone been smokers or non-smokers, I would recommend they word this sentence differently as the estimated risk ‘had everyone been a smoker/smoked 20 cigarettes per day’ vs. ‘had everyone been a non-smoker/smoked 0 cigarettes per day’ for the 11-year follow-up.

Our answer: Thank you. We revised the sentence according to the reviewer’s suggestion. 

4. In Tables 1 or 2, it would be helpful if authors presented the observed 11-year risk of CHD (ideally overall and by smoking status), as well as that simulated in the standard or mediational g-formula under no intervention. This would be a helpful assumption check, given the 'No intervention' output presented by the g-formula represents the simulated CHD risk under the Natural course (i.e. without intervention). If the parametric modelling process accounts for the causal model well, this 'no intervention' risk should be similar to that actually observed in the sample. If it is not, it may suggest more work is needed in refining the model specification, such as considering different covariates, interaction terms, or non-linear terms for continuous covariates.

Our answer: Thank you. The natural course compared to the observed risk was added to the discussion. In our study, the simulated 11-year risk of CHD under no intervention was 5.94% and the observed risk was 6.91%.

5. Either in Table 3 or separately, I would recommend authors present the simulated risk under each counterfactual exposure-mediator intervention. This would be helpful for readers, as it would allow multiplicative effects to be calculated such as E[YA1M1] / E[YA1M0] for the total IIE, as well as the interventional total direct effect and interventional pure indirect effect.

Our answer: thank you for your suggestion. They were added to table 3. 

Discussion:

Overall, authors do an excellent job of summarising related research as well as noting strengths and limitations of their study. Some brief areas of possible improvements are listed below:

1. While authors summarise findings of a number of related articles, I feel the findings of these studies could be more explicitly compared against their own findings, such as through simply restating the summary total, direct, and indirect effects of their own study before discussing total, direct, and weight-mediated effect estimates found in other studies.

Our answer. Thank you. Studies used different summary measures and different statistical models (adjusted for various covariates). However, their results indicated gaining weight after smoking cessation, and finally a potential small effect or no effect on CHD risk.

2. The discussion section could benefit from authors comparing effects of their analysis using the mediational g-formula to similar studies which did not utilise the mediational g-formula. For example, the 2013 JAMA study by Carole Clair et al. considers the mediating effect of weight in the relationship between smoking cessation and CVD among people with and without diabetes. This may be a more relevant comparison than the study by Nianogo and Arah where the outcome was type 2 diabetes mellitus and not directly related to CHD.

Our answer. The study of Nianogo and Arah was replaced with the results of 2013 JAMA study by Carole Clair et al. The list of references was revised as well. 

3. On page 11, authors suggest the parametric mediational g-formula is the only known method that is valid for investigating causal effects in the presence of time-varying mediation. This may not necessarily be correct, and could be adapted. As they note, VanderWeele and Tchetgen Tchetgen’s semiparametric approach is also valid. Further, the authors may find the 2017 paper by Zheng and van der Laan in the Journal of Causal Inference relevant, where they discuss the conditional mediation formula and multiple estimators for mediation effects in the presence of time-varying mediation through applying TMLE, IPW, or a non-targeted substitution estimator.

Our answer. Thank you. The comment about the parametric g-formula being the only known method was omitted.

4. Authors currently split their discussion of positivity, consistency, and exchangeability/unmeasured confounding into two sections, separated by study strengths. The discussion may be better laid-out if kept together, relating each assumption to the evidence supporting it and detracting from it one at a time.

Our answer. We bring these two parts together (one paragraph).

5. While authors appropriately note the g-null paradox as a limitation of the g-formula, if there is space in the manuscript it may benefit from a slightly longer discussion, especially given the authors note it is unlikely to be a concern in their study due to prior work suggesting the sharp null doesn’t hold in this context.

Our answer. Thank you. It was added. We made the following changes: Third, the parametric-formula is also subject to the g-null paradox—it could lead to rejecting the causal null, even when it is true.

6. While authors do an excellent job reviewing study strengths and limitations, they may wish to review work published in Statistics in Medicine in 2018 by Schomaker and Heumann, where they discuss combining bootstrap inference with multiple imputation where the g-formula is used. While this would not account for competing risks and censoring, and is not being recommended as necessary in this review, missing data such as that mentioned in point 2 of the ‘results’ improvement section can be accounted for as a step prior to or alongside the mediational g-formula, either through storing each bootstrap effect estimate across multiply imputed datasets and using percentile-based confidence intervals, or through applying the mediational g-formula to multiply imputed datasets and pooling summary effect estimates through Rubin's rules. Each approach discussed is relatively straight-forward to implement in practice.

Our answer: Thank you. Your suggestion was added to the discussion. We made the following changes: For future studies, Schomaker and Heumann (2018) regarding some estimators, which require bootstrapping to estimate confidence intervals, presented four methods to combine bootstrap estimation with multiple imputation. They indicated that the proportion of messiness and the number of multiple imputed data sets affect methods performance

---

## [Decision Letter · Decision Letter 1]

23 Dec 2021

Does Weight Mediate the Effect of Smoking on Coronary Heart Disease? Parametric Mediational G-Formula Analysis

PONE-D-21-18594R1

Dear Dr. Mansournia,

We’re pleased to inform you that your manuscript has been judged scientifically suitable for publication and will be formally accepted for publication once it meets all outstanding technical requirements.

Kind regards,

Y Zhan

Academic Editor

PLOS ONE

Additional Editor Comments (optional):

Reviewers' comments:

Reviewer's Responses to Questions

**Comments to the Author**

1. If the authors have adequately addressed your comments raised in a previous round of review and you feel that this manuscript is now acceptable for publication, you may indicate that here to bypass the “Comments to the Author” section, enter your conflict of interest statement in the “Confidential to Editor” section, and submit your "Accept" recommendation.

Reviewer #1: All comments have been addressed

Reviewer #2: All comments have been addressed

2. Is the manuscript technically sound, and do the data support the conclusions?

Reviewer #1: (No Response)

Reviewer #2: Yes

3. Has the statistical analysis been performed appropriately and rigorously? 

Reviewer #1: (No Response)

Reviewer #2: Yes

4. Have the authors made all data underlying the findings in their manuscript fully available?

Reviewer #1: (No Response)

Reviewer #2: Yes

5. Is the manuscript presented in an intelligible fashion and written in standard English?

Reviewer #1: (No Response)

Reviewer #2: Yes

6. Review Comments to the Author

Reviewer #1: (No Response)

Reviewer #2: (No Response)

7. PLOS authors have the option to publish the peer review history of their article (what does this mean?). If published, this will include your full peer review and any attached files.

Reviewer #1: No

Reviewer #2: **Yes: **Kieran Blaikie

---

## [Editor Report · Acceptance letter]

5 Jan 2022

PONE-D-21-18594R1 

Does Weight Mediate the Effect of Smoking on Coronary Heart Disease? Parametric Mediational G-Formula Analysis 

Dear Dr. Mansournia:

I'm pleased to inform you that your manuscript has been deemed suitable for publication in PLOS ONE. Congratulations! Your manuscript is now with our production department. 

Kind regards, 

on behalf of

Dr. Y Zhan 

Academic Editor

PLOS ONE